# Identifiability of deep generative models
# under mixture priors without auxiliary information

## Abstract

We prove identifiability of a broad class of deep latent variable models that (a) have universal approximation capabilities and (b) are the decoders of variational autoencoders that are commonly used in practice. Unlike existing work, our analysis does not require weak supervision, auxiliary information, or conditioning in the latent space. The models we consider are tightly connected with autoencoder architectures used in practice that leverage mixture priors in the latent space and ReLU/leaky-ReLU activations in the encoder. Our main result is an identifiability hierarchy that significantly generalizes previous work and exposes how different assumptions lead to different "strengths" of identifiability. For example, our weakest result establishes (unsupervised) identifiability up to an affine transformation, which already improves existing work. It's well known that these models have universal approximation capabilities and moreover, they have been extensively used in practice to learn representations of data.

## 1 INTRODUCTION

One of the key paradigm shifts in machine learning (ML) over the past decade has been the transition from hand-crafted features to automated, data-driven representation learning, typically via deep neural networks. One complication of automating this step in the ML pipeline is that it is difficult to provide guarantees on what features will (or won't) be learned. As these methods are being used in high stakes settings such as medicine, health care, law, and finance where accountability and transparency are not just desirable but often legally required, it has become necessary to place representation learning on a rigorous scientific footing. In order to do this, it is crucial to be able to discuss ideal, target features and the underlying representations that define these features. As a result, the ML literature has begun to move beyond consideration solely of downstream tasks (e.g. classification, prediction, sampling, etc.) in order to better understand the structural foundations of deep models.

Deep generative models (DGMs) such as variational autoencoders (VAEs) [Kingma and Welling, 2013, Rezende et al., 2014] are a prominent example of such a model, and are a powerful tool for unsupervised learning of latent representations, useful for a variety of downstream tasks such as sampling, prediction, classification, and clustering. Despite these successes, training DGMs is an intricate task: They are susceptible to posterior collapse and poor local minima [Yacoby et al., 2020, Dai et al., 2020, He et al., 2018, Wang et al., 2021], and characterizing their latent space remains a difficult problem [e.g. Klys et al., 2018, Van Den Oord et al., 2017]. For example, does the latent space represent semantically meaningful or practically useful features? Are the learned representations stable, or are they simply artifacts of peculiar choices of hyperparameters? These questions have been the subject of numerous studies in recent years [e.g. Schott et al., 2021, Luise et al., 2020, Locatello et al., 2019, Bansal et al., 2021, Csiszárik et al., 2021, Lenc and Vedaldi, 2015], and in order to better understand the behaviour of these models and address these questions, the machine learning literature has recently turned its attention to fundamental identifiability questions [Khemakhem et al., 2020a, D'Amour et al., 2020, Wang et al., 2021]. Identifiability is a crucial primitive in machine learning tasks that is useful for probing stability, consistency, and robustness. Without identifiability, the output of a model can be unstable and unreliable, in the sense that retraining under small perturbations of the data and/or hyperparameters may result in wildly different models.[1] In the context of deep generative models, the model output of interest is the latent space and the associated representations induced by the model.

---

[1]Formally, identifiability means the parametrization of the model is injective. See Section 2 for details.

*Submitted to the 38th Conference on Uncertainty in Artificial Intelligence* (UAI 2022). **To be used for reviewing only**.

In this paper, we revisit the identifiability problem in deep latent variable models and prove a surprising new result: Identifiability is possible under commonly adopted assumptions *and* without conditioning in the latent space, or equivalently, without weak supervision or side information in the form of auxiliary variables. This contrasts a recent line of work that has established fundamental new results regarding the identifiability of VAEs that requires conditioning on an auxiliary variable $u$ that renders each latent dimension conditionally independent [Khemakhem et al., 2020a]. While this result has been generalized and relaxed in several directions [Hälvä and Hyvarinen, 2020, Hälvä et al., 2021, Khemakhem et al., 2020b, Li et al., 2019, Mita et al., 2021, Sorrenson et al., 2019, Yang et al., 2021, Klindt et al., 2020, Brehmer et al., 2022], fundamentally these results still crucially rely on the side information $u$. We show that this is in fact unnecessary—confirming existing empirical studies [e.g Willetts and Paige, 2021, Falck et al., 2021]— and do so without sacrificing any representational capacity. What's more, the model we analyze is closely related to deep architectures that have been widely used in practice [Dilokthanakul et al., 2016, Falck et al., 2021, Jiang et al., 2016, Johnson et al., 2016, Lee et al., 2020, Li et al., 2018, Willetts et al., 2019, Lee et al., 2020]: We show that there is good reason for this, provide new insight into the properties of these models and support for their continued use.

**Overview** More specifically, we consider the following generative model for observations $x = (x_1, \ldots, x_n) \in \mathbb{R}^n$:

$$x = f(z) + \varepsilon \tag{1}$$

where the latent variable $z = (z_1, \ldots, z_m) \in \mathbb{R}^m$ follows a Gaussian mixture model (GMM), $f : \mathbb{R}^m \to \mathbb{R}^n$ is a piecewise affine nonlinearity such as a ReLU network, and $\varepsilon \in \mathbb{R}^n$ is independent, random noise.[2] We do not assume that the number of mixture components, nor the architecture of the ReLU network, are known in advance, nor do we assume that $z$ has independent components. Both the mixture model and neural network may be arbitrarily complex, and we allow for the discrete hidden state that generates the latent mixture prior to be high-dimensional and dependent. This includes classical ICA models (i.e. for which the latent variables are mutually independent) as a special case. Since both $z$ and $f$ are allowed to be arbitrarily complex, the model (1) has universal approximation capabilities, which is crucial for modern applications.

This model has been widely studied in the literature from a variety of different perspectives:

- *Nonlinear ICA.* When the $z_i$ are mutually independent, (1) recovers the standard nonlinear ICA model that has been extensively studied in the literature [Hyvärinen and Pajunen, 1999, Achard and Jutten, 2005, Zhang

---

[2]Our results include the noiseless case $\varepsilon = 0$ as a special case.

and Chan, 2008, Hyvarinen and Morioka, 2017, Hyvarinen et al., 2019, Hyvarinen and Morioka, 2016]. Although our most general results do not make independence assumptions, our results cover nonlinear ICA as a special case when $f$ is piecewise affine.

- *VAE with mixture priors.* When the prior over $z$ is a mixture model (e.g. such as a GMM), the model (1) is closely related to popular autoencoder architectures such as VaDE [Jiang et al., 2016], SVAE [Johnson et al., 2016], GMVAE [Dilokthanakul et al., 2016], DLGMM [Nalisnick et al., 2016], VampPrior [Tomczak and Welling, 2018], etc. Although such VAEs with mixture priors have been extensively studied, theoretical results are missing.

- *Warped mixtures.* Another closely related model is the warped mixture model of [Iwata et al., 2013], which is a Bayesian version of (1). Once again, theoretical guarantees for these models are lacking.

- *iVAE.* Finally, (1) is also the basis of the iVAE model introduced by [Khemakhem et al., 2020a], where identifiability (up to certain equivalences) is proved when there is an additional auxiliary variable $u$ that is observed such that $z_i \perp\!\!\!\perp z_j \,|\, u$.

**Contributions** Driven by this recent interest from both applied and theoretical perspectives, our main result (Theorem 3.5) shows that the model (1) is identifiable up to various *linear* equivalences, without conditioning or auxiliary information in the latent space. In fact, we develop a hierarchy of results under progressively stronger assumptions on the model, beginning with affine equivalence and ending up with a much stronger equivalence up to permutations only. Notably, under the weakest set of assumptions, our results already generalize existing work (Corollary 3.1) and answer an open question raised by Wang et al. [2021]. Full technical proofs are omitted and have been deferred to the full version of the paper.

**Related work** Classical results on nonlinear ICA [Hyvärinen and Pajunen, 1999] establish the nonidentifiability of the general model (i.e. without restrictions on $z$ and $f$); see also [Darmois, 1951, Jutten et al., 2003]. More recently, Khemakhem et al. [2020a] proved a major breakthrough by showing that given side information $u$, identifiability of the entire generative model is possible up to certain (nonlinear) equivalences. Since this pathbreaking work, many generalizations have been proposed [Hälvä and Hyvarinen, 2020, Hälvä et al., 2021, Khemakhem et al., 2020b, Li et al., 2019, Mita et al., 2021, Sorrenson et al., 2019, Yang et al., 2021, Klindt et al., 2020, Brehmer et al., 2022], all of which require some form of auxiliary information. Other approaches to identifiability include various forms of weak supervision such as contrastive learning [Zimmermann et al., 2021], group-based disentanglement [Locatello et al., 2020],

and independent mechanisms [Gresele et al., 2021]. Non-identifiability has also been singled out as a contributing factor to practical issues such as posterior collapse in VAEs [Wang et al., 2021, Yacoby et al., 2020].

Our approach is to avoid additional forms of supervision altogether, and enforce identifiability in a purely unsupervised fashion. Recent work along these lines includes [Wang et al., 2021], who propose to use Brenier maps and input convex neural networks, and [Moran et al., 2021] who leverage sparsity and an anchor feature assumption. Aside from different assumptions, the main difference between this line of work and our work is that their work only identifies the latent space $P(Z)$, whereas our focus is on jointly identifying *both* $P(Z)$ and $f$. In fact, we provide a decoupled set of assumptions that allow $f$ or $P(Z)$ or both to be identified. Thus, we partially resolve in the affirmative an open problem regarding model identifiability raised by the authors in their discussion.

Another distinction between this line of work and the current work is our focus on architectures and modeling assumptions that are *standard* in the deep generative modeling literature, specifically ReLU nonlinearities and mixture priors. As noted above, there is a recent tradition of training variational autoencoders with mixture priors [Dilokthanakul et al., 2016, Falck et al., 2021, Jiang et al., 2016, Johnson et al., 2016, Lee et al., 2020, Li et al., 2018, Willetts et al., 2019, Lee et al., 2020]. Our work builds upon this empirical literature, showing that there is good reason to study such models: Not only have they been shown to be more effective compared to vanilla VAEs, we show that they have appealing theoretical properties as well. In fact, recent work [Willetts and Paige, 2021, Falck et al., 2021] has observed precisely the identifiability phenomena studied in our paper, however, this work lacks rigorous theoretical results to explain these observations.

## 2 PRELIMINARIES

We first introduce the main generative model that we study and its properties, and then proceed with a brief review of identifiability in deep generative models.

**Generative model**  The observations $x \in \mathbb{R}^n$ are realizations of a random vector $X$, and are generated according to the generative model (1), where $z \in \mathbb{R}^m$ represents realizations of an unobserved random vector $Z$. We make the following assumptions on $Z$ and $f$:[3]

(P1)  $P(Z)$ is a Gaussian mixture model with an unknown

---

number of components, i.e., for $\lambda_j > 0$

$$p(z) = \sum_{j=1}^{J} \lambda_j \varphi(z; \mu_j, \Sigma_j), \quad \sum_{j=1}^{J} \lambda_j = 1, \quad (2)$$

where $p(z)$ is the density of $P(Z)$ with respect to some base measure, and $\varphi(z; \mu_j, \Sigma_j)$ is the gaussian density with mean $\mu_j$ and covariance $\Sigma_j$.

(F1)  $f$ is a piecewise affine function, such as a multilayer perceptron with ReLU (or leaky ReLU) activations.

Recall that an affine function is a function $x \mapsto Ax + b$ for some matrix $A$. As already discussed, special cases of this model have been extensively studied in both applications and theory, and both (P1)-(F1) are quite standard in the literature on deep generative models and represent a useful model that is widely used in practice [e.g. Dilokthanakul et al., 2016, Falck et al., 2021, Jiang et al., 2016, Johnson et al., 2016, Lee et al., 2020, Li et al., 2018, Willetts et al., 2019, Lee et al., 2020]. In particular, when $J = 1$ this is simply a classical VAE with an isotropic Gaussian prior.

**Identifiability**  A statistical model is specified by a (possibly infinite-dimensional, as in our setting) parameter space $\Theta$, a family of distributions $\mathcal{P}$, and a mapping $\pi : \Theta \to \mathcal{P}$; i.e. $\pi(\theta) \in \mathcal{P}$ for each $\theta \in \Theta$. In more conventional notation, we define $\mathcal{P} = \{p_\theta : \theta \in \Theta\}$, in which case $p_\theta = \pi(\theta)$. A statistical model is called *identifiable* if the parameter mapping $\pi$ is one-to-one (injective). In practical applications, the strict definition of identifiability is too strong, and relaxed notions of identifiability are sufficient. Classical examples include identifiability up to permutation, re-scaling, or orthogonal transformation. More generally, a statistical model is *identifiable up to an equivalence relation* $\sim$ defined on $\Theta$ if $\pi(\theta) = \pi(\theta') \implies \theta \sim \theta'$. For more details on the different notions of identifiability in deep generative models, see Khemakhem et al. [2020a,b], Roeder et al. [2021].

More precisely, we use the following definition. Let $f_\sharp P$ denote the pushforward measure of $P$ by $f$.

**Definition 2.1.** Let $\mathcal{P}$ be a family of probability distributions on $\mathbb{R}^m$ and $\mathcal{F}$ be a family of functions $f : \mathbb{R}^m \to \mathbb{R}^n$.

1. For $(P, f) \in \mathcal{P} \times \mathcal{F}$ we say that $P$ is *identifiable (from $f_\sharp P$) up to an affine transformation* if for any $(P', f') \in \mathcal{P} \times \mathcal{F}$ s.t. $f_\sharp P \equiv f'_\sharp P'$ there exists an invertible affine map $h : \mathbb{R}^m \to \mathbb{R}^m$ such that $P' = h_\sharp P$ (i.e., $P'$ is the pushforward measure of $P$ by $h$).

2. If there exists such $h$, such that $f' = f \circ h^{-1}$ and $P' = h_\sharp P$, we say that $(P, f)$ is *identifiable (from $f(P)$) up to an affine transformation*.

This definition can be extended to transformations besides affine transformations (e.g. permutations, translations, etc.) in the obvious way.

---

[3]In the sequel, we will use (P#) to index assumptions on the prior $P(Z)$, and (F#) to index assumptions on the decoder $f$.

Identifiability is a crucial property for a statistical model: Without identifiability, different training runs may lead to very different parameters, making training unpredictable and replication difficult. The failure of identifiability, also known as *underspecification* and *ill-posedness*, has recently been flagged in the ML literature as a root cause of many failure modes that arise in practice [D'Amour et al., 2020, Yacoby et al., 2020, Wang et al., 2021]. As a result, there has been a growing emphasis on identification in the deep learning literature, which motivates the current work. Finally, in addition to these reproducibility and interpretability concerns, identifiability is a key component in many applications of latent variable models including causal representation learning [Schölkopf et al., 2021], independent component analysis [Comon, 1994], and topic modeling [Arora et al., 2012, Anandkumar et al., 2013]. See [Ran and Hu, 2017] for additional discussion and examples.

**Auxiliary information and iVAE**  It is well-known that assuming independence of the latent factors—i.e. $Z_i \perp\!\!\!\perp Z_j$—is insufficient for identifiability [Locatello et al., 2019]. Recent work, starting with iVAE, shows identifiability by additionally assuming that a $k$-dimensional auxiliary variable $u$ is observed such that $p(z \mid u)$ is conditionally factorial, i.e. $Z_i \perp\!\!\!\perp Z_j \mid U$. This extra information serves to break symmetries in the latent space and is crucial to existing proofs of identifiability.

To make the connection with this work clear, observe that assumption (P1) is equivalent to assuming that there is a discrete hidden state $U \in \{1, \ldots, J\}$ such that $P(Z = z \mid U = j) = p_j(z)$ and $P(U = j) = \lambda_j$. More generally, $U = (U_1, \ldots, U_k)$ may be multivariate. In this way, a direct parallel between our work and previous work is evident, with several crucial caveats:

- We do *not* assume that $U$ is observed—even partially—or known in any way;

- We allow for the $Z_i$ to be arbtrarily dependent even after conditioning on $U$, and this dependence need not be known;

- We do not even require the number of states $J$ to be known, and we do not require any bounds on $J$ (e.g. iVAE requires $J \geq m + 1$).

- In the case where $U$ is multivariate (i.e $k := \dim(U) > 1$), we do not require the number of latent dimensions $k$, the state spaces, or their dependencies to be known.

In order to break the symmetry without knowing anything about $U$ or its dependencies, we must develop fundamentally new insights into nonparametric identifiability of latent variable models.

# 3   MAIN RESULTS

For any positive integer $d$, let $[d] = \{1, \ldots, d\}$. By (P1), we can write the model (1) as follows. Let $U = (U_1, \ldots, U_k) \in [d_1] \times \cdots \times [d_k]$ where $d_i := \dim(U_i)$ and $k := \dim(U)$; we allow $U$ to be multivariate ($k > 1$) and dependent—i.e., we do not assume that the $U_i$ are marginally independent. It follows trivially from (P1) that $P(U_1 = u_1, \ldots, U_k = u_k) \in \{\lambda_1, \ldots, \lambda_J\}$ and $J = \prod_i d_i$, where we recall that $J$ is the *unknown* number of mixture components in $P(Z)$. Denote the marginal distribution of $U$, which depends on $\lambda_j$, by $P_\lambda$. The variables $(U, Z)$ are unobserved and encode the underlying latent structure:

$$
\left.
\begin{aligned}
U = u &\sim P_\lambda(U = u) \\
[Z \mid U = u] &\sim N(\mu_u, \Sigma_u) \\
[X \mid Z = z] &\sim f(z) + \varepsilon,
\end{aligned}
\right\} \implies U \to Z \to X. \quad (3)
$$

Here, $P_\lambda$ is the distribution on $U$ described above. Our goal is to identify the latent distribution $P(U, Z)$ and/or the nonlinear decoder $f$ from the marginal distribution $P(X)$ induced by (3).

Our main result, Theorem 3.5, provide a hierarchy of progressively stronger conditions under which $P(U, Z)$, $f$, or both, can be identified in progressively stronger ways. The idea is to illustrate explicitly what conditions are sufficient to identify the latent structure up to the corresponding equivalence relation.

We defer the statement of the main results to Section 3.3, after the main conditions have been described. As a preview to the main results, we first present the following corollary:

**Corollary 3.1.** *Suppose $k = \dim(U) = 1$, $J \geq 1$, $(U, Z)$ are unobserved, and $X$ is observed. (a) If $f$ is an invertible ReLU network, then both $P(U, Z)$ and $f$ are identifiable up to an affine transformation. (b) If $f$ is only weakly injective, then $P(U, Z)$ is identifiable up to an affine transformation.*

For comparison, Corollary 3.1 already strengthens existing results, since $U$ is not required to be known and we are able to identify $f$. In fact, the latter answers an open question raised by [Wang et al., 2021]. What's more, this is just the *weakest* result implied by our main results: Under stronger assumptions on the latent structure, the affine equivalence presented above can be strengthened further.

Taken together, the results in this section have the following concrete implication for practitioners: For stably training variational autoencoders, there is now compelling justification to work with a GMM prior and deep ReLU/Leaky-ReLU networks. As we saw above, this is commonly done in practice already.

## 3.1 POSSIBLE ASSUMPTIONS ON $f$

To distinguish cases where $f$ is and is not identifiable, we require the following technical definition. Recall that for sets $A, B$, $f^{-1}(A) = \{x : f(x) \in A\}$ and $f(B) = \{f(x) : x \in B\}$. A function need not be invertible for the preimage $f^{-1}(A)$ to be well-defined.

**Definition 3.2.** Let $m \leq n$ and $f : \mathbb{R}^m \to \mathbb{R}^n$.

(F2) We say that $f$ is *weakly injective* if (i) there exists $x_0 \in \mathbb{R}^n$ and $\delta > 0$ s.t. $|f^{-1}(\{x\})| = 1$ for every $x \in B(x_0, \delta) \cap f(\mathbb{R}^m)$, and (ii) $\{x \in \mathbb{R}^n : |f^{-1}(\{x\})| = \infty\} \subseteq f(\mathbb{R}^m)$ has measure zero with respect to the Lebesgue measure on $f(\mathbb{R}^m)$.

(F3) We say that $f$ is *injective* if $|f^{-1}(\{x\})| = 1$ for every $x \in f(\mathbb{R}^m)$.

**Example 3.3.** For example, although $x \mapsto \mathrm{ReLU}(x)$ is not injective, it is weakly injective, where $\mathrm{ReLU}(x) = \max\{0, x\}$ is the usual rectified linear unit. To see this, note that image of ReLU is the set $\mathbb{R}_{\geq} = \{y \mid y \geq 0\}$, and ReLU has the unique preimage for every $y \in \mathbb{R}_{>} = \{y \mid y > 0\}$. Clearly, $(\mathbb{R}_{\geq} \setminus \mathbb{R}_{>}) = \{0\}$ has measure zero inside $\mathbb{R}_{\geq}$. Under simple assumptions on the architecture of a RelU network, it is generically weakly injective. At the same time, $x \mapsto 0$ and $x \mapsto |x|$ are not weakly injective.

In general, a deep ReLU network may be even injective, e.g. $\mathrm{ReLU}(x) + \mathrm{ReLU}(-x) = x$ for $x \in \mathbb{R}$).

## 3.2 POSSIBLE ASSUMPTIONS ON $Z$

Our weakest result requires no additional assumptions on $Z$ beyond (P1); see Corollary 3.1. Under stronger assumptions, more can be concluded. As with the previous section, the assumptions presented here are not necessary, but may be imposed in order to extract stronger results.

The first condition is a mild condition that allows us to strengthen affine identifiability:

(P2) $Z_i \perp\!\!\!\perp Z_j \mid U$ for all $i \neq j$ and there exist a pair of states $U = u_1$ and $U = u_2$ such that all $((\Sigma_{u_1})_{tt} / (\Sigma_{u_2})_{tt} \mid t \in [m])$ are distinct.

The second condition is more technical, and is only necessary if $k > 1$ and we wish to identify $P(U)$ in addition to $P(Z)$. In fact, not only will we recover $P(U)$, but also the (unknown) number of hidden variables (i.e. $k$) and their state spaces (i.e. $d_j$). Note that $P(U)$ is not needed to sample from (1), as long as we have $P(Z)$. Before introducing this condition, we need a preliminary definition.

**Definition 3.4.** Let $U_{-i}$ denote $\{U_j : j \neq i\}$. We define $\mathrm{ne}(U_i) = [m] \setminus \{t : Z_t \perp\!\!\!\perp U_i \mid U_{-i}\}$ and $\mathrm{ne}(Z_i) = \{t : Z_i \in \mathrm{ne}(U_t)\}$. For a subset $Z' \subset Z$, $\mathrm{ne}(Z') = \cup_{Z_i \in Z'} \mathrm{ne}(Z_i)$.

The neighborhood $\mathrm{ne}(U_i)$ collects the variables $Z_t$ that depend on $U_i$ directly.

(P3) The following conditions hold:
(a) For all $Z' \subset Z$ and $u_1 \neq u_2$, $P(Z' \mid \mathrm{ne}(Z') = u_1) \neq P(Z' \mid \mathrm{ne}(Z') = u_2)$;
(b) If $P(U', Z, X) = P(U, Z, X)$, then $\dim(U') \leq \dim(U)$;
(c) For any $U_i \neq U_j$, $\mathrm{ne}(U_i)$ is not a subset of $\mathrm{ne}(U_j)$.

Condition (P3) is a "maximality" condition that is adapted from [Kivva et al., 2021]: We are interested in identifying the most complex latent structure with the most number of hidden variables. This is in fact necessary since we can always merge two (or more) hidden variables into a single hidden variable without changing the joint distribution.

## 3.3 MAIN IDENTIFIABILITY RESULT

When $\dim(U) = 1$, there is no additional structure in $U$ to learn, and so the setting simplifies considerably. In particular, learning $P(Z)$ is equivalent to $P(U, Z)$.

The case $\dim(U) > 1$ is especially challenging: Unlike previous work such as iVAE that assumes $U$ (and hence its structure) to be known, we do not assume anything about $U$ is known. Thus, everything about $U$ must be reconstructed based on $P(X)$ alone, hence the need for (P3) to identify $P(U)$ below. Note, when $\dim(U) = 1$, (P3) holds.

**Theorem 3.5.** *Under (P1)-(F1), we have the following:*

(a) *(F2) $\implies$ $P(Z)$ is identifiable from $P(X)$ up to an affine transformation.*

(b) *(F2)+(P2) $\implies$ $P(Z)$ is identifiable from $P(X)$ up to permutation, scaling, and/or translation.*

(c) *(F2)+(P2)+(P3) $\implies$ $(k, d_1, \ldots, d_k, P(U, Z))$ are identifiable from $P(X)$ up to a permutation of $U$, and permutation, scaling, and/or translation of $Z$.*

(d) *If additionally (F3) holds, and $f$ is continuous, then $f$ is also identifiable from $P(X)$ up to an affine transformation.*

If the assumption (F2) ($f$ is weakly injective) is removed, then the claim of Theorem 3.5 is not true.

## 4 EXPERIMENTS

There has been extensive work already to verify empirically that the model (1) under (P1)-(F1) is identifiable. For example, [Willetts and Paige, 2021] observe that deep generative models with clustered latent spaces are empirically identifiable, and compared this directly to models that rely on side information, and [Falck et al., 2021] show that meaningful latent variables can be learned consistently in a fully

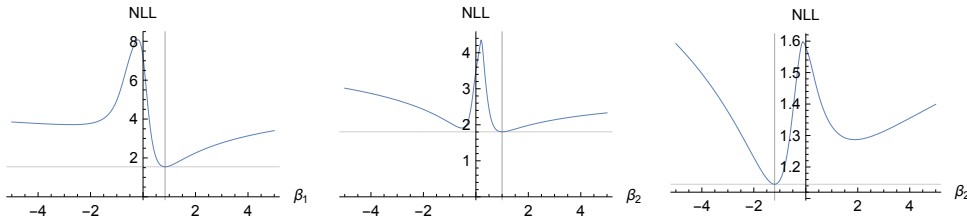

Figure 1: Selected examples of negative log-likelihood for different runs. Vertical lines indicate the ground truth and (global) minimizer, which always coincide.

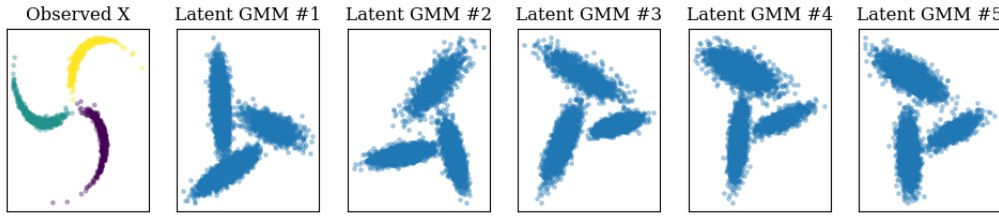

Figure 2: Recovered latent spaces for 5 runs of VaDE on pinwheel dataset with 3 clusters

unsupervised manner even when $U$ has high-dimensional structure. Moreover, [Falck et al., 2021] indicate that high-dimensional structure is important for improved performance. Beyond these, it is well-known that VAEs with mixture priors such as VaDE [Jiang et al., 2016] achieve competitive performance on many benchmark tasks; see [Dilokthanakul et al., 2016, Falck et al., 2021, Johnson et al., 2016, Lee et al., 2020, Li et al., 2018, Willetts et al., 2019, Lee et al., 2020] for additional experiments and verification. Building upon the established success of these methods, we augment these experiments as follows: 1) We use simple examples to verify that the likelihood indeed has a unique minimizer at the ground truth parameters; 2) We train VaDE on (misspecified) simulated toy models; and 3) We measure stability (up to affine transformations) of the learnt latent spaces on real data. To measure this, we report the Mean Correlation Coefficient [Khemakhem et al., 2020b, Appendix A.2] metric, which is standard.

**Maximum likelihood** We simulated models satisfying (P1)-(F1) by randomly choosing weights and biases for a single-layer ReLU network and randomly generating a GMM with $J = 2$ or 3 components. These models are simple enough that exact computation of the MLE along with the likelihood surface is feasible via numerical integration (Figure 1). In all our simulations (50 total), the ground truth was the unique minimizer of the negative log-likelihood, as predicted by the theory. These examples also illustrate a small-scale test of misspecification in the theoretical model: We include cases where $J$ is misspecified and $f$ fails to satisfy (F3), but the MLE succeeds anyway.

**Simulated data** In our experiments on synthetic datasets we fit VaDE to observed data 5 times. Let $Z^{(1)}, \ldots, Z^{(5)}$

be the learned latent spaces. For every pair $Z^{(i)}, Z^{(j)}$ we evaluate the MCC For instance, for the pinwheel dataset with three clusters the average weak MCC is 0.87 and the average strong MCC is 1.0. This shows strong evidence of recovery of the latent space up to affine transformations.

**Real data** We measure stability of the learnt latent space by training MFCVAE Falck et al. [2021] on MNIST 10 times with different initializations and then comparing the latent representations learnt. The strong MCCs are computed to be 0.7 (ReLU), 0.69 (LeakyReLU) and the weak MCCs are computed to be 0.91 (ReLU), 0.94 (LeakyReLU). These observations validate the observations first made in Willetts and Paige [2021], who ran extensive experiments on VaDE and iVAE on several large datasets including MNIST, SVHN and CIFAR10. These strong correlations confirm our theory and are of particular importance to practitioners for whom stability of learning is of the essence.

## 5 CONCLUSION

We have a proved a general series of results describing a hierarchy of identifiability for deep generative models that are currently used in practice. Our experiments confirm both on exact and approximate simulations that identifiability indeed holds in practice. An obvious direction for future work is to study finite-sample identifiability problems such as sample complexity and robustness (i.e. how many samples are needed to ensure that the global minimizer of the likelihood is reliably the ground truth?). It is an important open question to use these insights to develop better algorithms and optimization techniques that work on finite-samples with misspecified models (i.e. real data).

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
