# OpenReview forum: "Identifiability of deep generative models under mixture priors without auxiliary information"
_auai.org/UAI/2022/Workshop/CRL — CRL@UAI 2022 Poster_

### Official Review · Reviewer_ennC · 2022-06-28
**Identifiability without non-IID data.**

**Rating:** 7
**Confidence:** 3

**Review:**

I thought this is an interesting paper that was well-written and deserves to be discussed at the workshop, but I do have a number of reservations with their assumptions. The authors consider the standard representation learning setup of Z \sim P(Z) and X = f(X) show that we can rely on assumptions on parametric assumptions on f(.) and P(.) to attain identification rather than using conditional independence on the latents or some form of non-IDD data. The proofs aren't included, but the results seem likely to hold and are supported by simple experiments. My main concern is with the assumptions on f(.). They assume:
1. f(.) is piecewise affine (as in a ReLU network).
2. Assumption F2 assumes that f(.) is weakly injective (see paper for details - it's essentially assuming that f(.) is invertible *almost* everywhere in its image).

Assumption 1 is weird given that it's usually applied to the map from latents to pixels - which is definitely not a ReLU network. It's a camera or a rendering engine or some similar "function" that takes a state of the world and renders our pixels. I'm guessing that the authors think of this as approximable via a piecewise affine function which is more plausible given the recent success in generative models, but I still think that gap should be discussed more.

Assumption 2 seems to assume away the hard problem of a parametric class and I'm not sure how you'd check it. The authors claim that this holds under simple assumptions on a ReLU network that parameterizes f(.) and maybe that's true - but generally we're not in control of f(.) (because f is a camera or a rendering engine). In a real problem, I don't see how you'd know whether or not your problem is identified?

---

### Official Review · Reviewer_xrrg · 2022-06-29

**Rating:** 7
**Confidence:** 4

**Review:**

Summary:

This paper considers identifiability of deep generative models:
x = f(z) + epsilon
where x are the observed data, z are the factors and epsilon is independent noise. The factors z are assumed to be distributed as a Gaussian mixture model. The mapping f is assumed to be a piecewise affine nonlinearity such as a ReLU network.

The paper proves:
1. If f is weakly injective, p(z) is identifiable up to affine transformation.
2. If (i) f is weakly injective, (i) z_i is conditionally independent of z_j (given U, the mixture categorical variable) and (iii) the latent Gaussian variances for different mixtures are distinct, then p(z) is identifiable up to permutation, scaling and translation.
3. If, in addition to conditions in 2, a "maximality condition" on the mixture model holds, then the number of mixture components is identifiable, in addition to p(z, u), up to permutation, scaling and translation.
4. If f is injective, then p(z) and f are identifiable up to affine transformation.

The paper conducts experiments on VaDE (a mixture model by Jiang et al. 2017). They show the replicability of VaDE's results over different runs on (i) synthetic data and (ii) MNIST.

Pros:
- Identification of VAEs with Gaussian mixture model factor distributions is new and an important contribution.
- The paper is well written and organized.

Cons:
- The proofs are not included so it is hard to verify results.
- The abstract does not mention the requirement of a Gaussian mixture model distribution on the factors, nor the conditional independence condition on the factors.
- I would like to see more discussion of the intuition behind assumptions P2, P3, and when they are expected to hold.

A more general comment is I would like to understand is the limitations of the GMM assumption on the factors. It seems like if z are very separated, then this separation is preserved in x and so it can be identified. However, generally disentanglement is related to how the coordinates of z are mixed to reconstruct x = f(z). That is, often the clusters are within the coordinates of z, not over the entire z vectors. When can we expect the entire z vector to be cluster-able e.g. in a Gaussian mixture?

---

### Meta-Review · Program_Chairs · 2022-07-06

**Recommendation:** Accept (Poster)
**Confidence:** 4

**Metareview:**

Both reviewers agree that the paper presents new and interesting identifiability results. The authors are encouraged to address the reviewers' questions and comments for the camera ready version of the paper.

---

### Decision · Program_Chairs · 2022-07-06

Accept (Poster)